# *Nopalea cochenillifera* Biomass as Bioadsorbent in Water Purification

**Vitória Régia do Nascimento Lima [1], Álvaro Gustavo Ferreira da Silva [1], Renata Ranielly Pedroza Cruz [2,\*], Luana da Silva Barbosa [3], Neilier Rodrigues da Silva Junior [4], Giuliana Naiara Barros Sales [1], Marcelo Augusto Rocha Limão [2], Franciscleudo Bezerra da Costa [1], Pahlevi Augusto de Souza [5], Kilson Pinheiro Lopes [1], Riselane de Lucena Alcântara Bruno [3], Alberício Pereira de Andrade [6] and Wellington Souto Ribeiro [2]**

[1] Centro de Ciências e Tecnologia Agroalimentar, Programa de Pós-Graduação em Horticultura Tropical, Universidade Federal de Campina Grande, Pombal 58840-000, Brazil; vitoriatenps@hotmail.com (V.R.d.N.L.); alvarogustavosilva@gmail.com (Á.G.F.d.S.); giulianasales@outlook.com (G.N.B.S.); franciscleudo@yahoo.com.br (F.B.d.C.); kilsonlopes@gmail.com (K.P.L.)

[2] Departamento de Agronomia, Programa de Pós-Graduação em Fitotecnia, Universidade Federal de Viçosa, Viçosa 36570-900, Brazil; marcelo.limao@ufv.br (M.A.R.L.); wellington.souto@ufv.br (W.S.R.)

[3] Departamento de Fitotecnia de Ciências Ambientais, Programa de Pós-graduação em Agronomia, Universidade Federal da Paraíba, Areia 58397-000, Brazil; luanabarbosassb@gmail.com (L.d.S.B.); lane@cca.ufpb.br (R.d.L.A.B.)

[4] Programa de Pós-Graduação em Bioquímica Aplicada, Universidade Federal de Viçosa, Viçosa 36570-900, Brazil; neilierjunior@hotmail.com

[5] Instituto Federal de Educação, Ciência e Tecnologia do Rio Grande do Norte, Campus Currais Novos, Currais Novos 59380-000, Brazil; pahlevi.souza@ifrn.edu.br

[6] Universidade Federal do Agreste Pernambucano, Garanhus 55292-270, Brazil; albericio3@gmail.com

\* Correspondence: renata.pedroza@ufv.br

**Abstract:** Contaminated water consumption is one of the greatest risks to human health, especially in underdeveloped and developing countries. Water is a universal right, but millions of people worldwide consume untreated surface water. The objective in this study is to evaluate water purification with *Nopalea cochenillifera* var. Miúda biomass. Fragments (1, 2, and 3 g) of *N. cochenillifera* were added to the aqueous solution containing red-yellow Chromic and Podzolic Luvisol simulating turbid water sources in Northeastern Brazil. The total, non-structural (i.e., reducing and non-reducing sugars, alcohol insoluble solids), and structural (i.e., pectin) carbohydrates, adsorption kinetics, turbidity, electrical conductivity, pH, zeta potential, and total coliforms presence were evaluated. Findings show that the *Nopalea cochenillifera* biomass adsorbed the suspended particles in the aqueous solution, making it more translucent due to the complex and heterogeneous adsorbents' ion exchange matrix, but the biomass addition did not eliminate total coliforms from the aqueous solution. We concluded that the *Nopalea cochenillifera* biomass water treatment reduces suspended dissolved particles and turbidity, but it needs to be associated with other treatments to eliminate total coliforms and ensure water safety for human consumption.

**Keywords:** ion exchange biological matrix; polyelectrolytes; water treatment; turbidity

## 1. Introduction

Safe and readily available water is important for public health, home use, food production, and recreational purposes [1]. In 2010, the UN General Assembly recognized the human right to water and sanitation [2]. It is agreed that all individuals have the right to sufficient, continuous, safe, acceptable, and physically accessible water for personal and domestic use [3]. For this, water must be available in the environment in adequate quantity and quality. For human consumption, water must be odorless, colorless, and tasteless, without harmful chemicals and microorganisms [4].

In 2015, 423 million people worldwide used water from unprotected wells and springs, and 159 million consumed untreated surface water from lakes, lagoons, rivers, and streams [1]. These untreated water sources can transmit cholera, diarrhea, hepatitis A, typhoid, and polio [5]. A lack of inadequate or inadequately managed water treatment and sanitation exposes the population to avoidable risks [6].

In Brazil, contaminated water consumption accounts for two thirds of the deaths of children under five in public hospitals with 240 deaths per day [7]. This situation is most severe in the drought polygon with 1348 municipalities in the states of Alagoas, Bahia, Ceará, Minas Gerais, Paraíba, Pernambuco, Piauí, Rio Grande do Norte, and Sergipe [8]. Water from reservoirs, barriers, and cisterns in Brazilian semiarid regions is typically supplied by water trucks and does not undergo cleaning or disinfection, thus making it vulnerable to contamination by bacteria, protozoa, and heavy metals.

Drought is a phenomenon that limits water quality and quantity where it occurs. The drinking water deficit in Brazil is 30%, harming over 40 million people [9]. This, coupled with a lack of public policies, threatens health and makes human life vulnerable. Resources for adaptation or living with drought are not equally distributed, and poor populations are usually disadvantaged [10].

In the drought polygon, as in other worldwide areas with similar characteristics, the drought periods' unpredictability, population growth, demographic change, and urbanization increase the challenge for water supply [11–13]. It is estimated that by 2025, half of the world's population will be living in areas lacking water [1,5].

The levels of the largest reservoirs in Northeastern Brazil have not exceeded 32% of their maximum capacity in the last seven years. In addition to the negative outlook for the energy sector, water tanks for human consumption are also in a similar situation— completely dry or dead volume. For the few reservoirs that do not dry up, they store turbid water, with high suspended organic matter content and the potential for microbiological contamination [14]. This same water, however, is used for the consumption of many families and animals and may aggravate infectious and parasitic disease transmission [15].

To provide quality water supply to the population and for their daily activities, there is water treatment using metallic salts. This consists of a unit operations series, including coagulation, a process that removes chemicals in solution and destabilizes colloidal suspensions and solids that cannot be removed by sedimentation or filtration [16]. Flocculation is closely linked to coagulation, during which the particles are destabilized by the coagulant agglutinate and form settling flakes [17]. However, the use of metallic salts for water treatment has relatively high acquisition costs for developing countries, and it also involves the production of large volumes of sludge, significant changes in the pH of treated water, and residues that can harm human health and the environment [18]. Only 89% of the world population (6.5 billion people) have access to treated water [1], and the disadvantages of using metallic salts can be replaced by natural polymers derived from plant material to produce the coagulation/flocculation process.

Natural polymers of plant origin, accessible to all, can adsorb solid contaminant particles with characteristics similar to chemical adsorbents [17,19] and can be an emergency alternative for a population without access to treated water. These biopolymers should be biodegradable, easily accessible, and inexpensive, and they should involve easy operation, have a lower sensitivity to toxic loadings, and not alter the pH of the treated water. Recently, biopolymers from different plants have been studied [20,21]. These have chemical structures with many hydroxyphenolic groups that give them the property of forming complexes with many macromolecules, such as proteins, hydrated carbons, or metal ions [22]. Cacti use in water treatment was recently compared to other natural coagulants such as *Moringa oleifera* Lam. (Brassicales: Moringaceae), *Strychnos potatorum* L.f. (Gentianales: Loganiaceae), *Melocactus* sp. L. (Caryophyllales: Cactaceae), *Opuntia dillenii* (Ker Gawl.) Haw. (Caryophyllales: Cactaceae), *Stenocereus griseus* (Haw.) Buxb. (Caryophyllales: Cactaceae), *Cereus forbesii* L. (Caryophyllales: Cactaceae), *Aloe arborescens* Mill. (Asparagales:

Asphodelaceae), *A. vera* L. Burm. f. (Asparagales: Asphodelaceae), and *Cicer arietinum* L. (Fabales: Fabaceae) [23–25].

Cactus is an important forage option for the world's semi-arid regions. The energy and water provided by this forage has long been used for livestock, and cacti cultivation has increased food security and reduced hunger and poverty in Brazil's semi-arid areas and around the world. Environmental and management practices are minimal for cacti, whose production is economically viable for the population of this region. The most cultivated cactus genera in Northeastern Brazil are *Nopalea* and *Opuntia*, and the main species are *O. ficus-indica* Mill., *O. stricta* (Haw.) Haw., and *N. cochenillifera* Salm Dyck. Other species that are also used include *O. lindheimeri* Engelm, *O. ellisiana*, *O. engelmannii* Salm Dyck, *O. chrysacantha* Berg, *O. amyclae*, and *O. rastrera* Weber. In Northeastern Brazil, the *N. cochenillifera* cultivation predominates due to higher concentrations of dry matter, low nutrient demand, and drought tolerance [26].

The adsorption properties of various low-cost bioadsorbents with plant biomass, agricultural residues, and their activated carbons have been reported in recent years. These bioadsorbents can be chemically modified to improve their efficiency and can be reused to extend their applicability on an industrial scale [24]. With this in mind, the objective of this work is to evaluate water purification using *N. cochenillifera biomass.*

## 2. Materials and Methods

### 2.1. Experiment Design

The experiment was carried out in a completely randomized design with three treatments containing 1, 2, and 3 g of *N. cochenillifera* biomass and 10 replications. The experiment was carried out in triplicate.

### 2.2. Preparation of Cloudy Aqueous Solution

Surface horizon samples of Chromic Luvisol [25] were collected from Pombal, Paraíba, Brazil (Sertão) and from red yellow podzolic (MA/EPE—SUDENE/DRN, 1972) in Lagoa Seca, Paraíba, Brazil (Agreste). Soil samples were dried and sieved (Bronzinox Mesh 6 aperture 3.36 mm 8 × 2′). Fifty grams of sieved soil was added to 500 mL of distilled water. The solution was stirred clockwise and counterclockwise on a shaking table for 1 h (130 rpm) until clay particles' uniform dispersion and was allowed to stand for 24 h for complete particles hydration. The supernatant was used as a stock aqueous solution for simulating the water used for human and other animal watering in Northeastern Brazil.

### 2.3. Origin and Biomass Preparation

*Nopalea cochenillifera* var. Miúda were taken from third order rackets harvested at the experimental farm of Professor Rolando Enrique Rivas Castellón of the Universidade Federal de Campina Grande, Paraíba, Brazil (6°48′45″ south latitude and 37°55′43″ west longitude). The culture was three years old. The predominant dovecote climate is BSh (Köppen) type, warm semiarid, with average annual precipitation of 750 mm, and concentrated rainfall from December to April. Temperature and relative humidity were 36 ± 5 °C and 20 ± 2%, respectively. The Lagoa Seca predominant climate is Cfb (Köppen), warm and temperate, with an average annual rainfall of 1760 mm distributed throughout the year. Temperature and relative humidity were 17 ± 5 °C and 85 ± 2%, respectively. Fresh *N. cochenillifera* cladodes were washed with tap water and plenty of distilled water to remove dirt and dust and were cut into 1, 2, and 3 g fragments, without peeling.

### 2.4. Experimental Procedure and Evaluations

The aqueous solutions were stirred again and after one hour of rest, then fragments of 1, 2, and 3 g of *N. cochenillifera* biomass were added. Quality assessments of the aqueous solution were made two hours after the addition of *N. cochenillifera* fragments.

### 2.4.1. Alcoholic Extracts

Five grams of *N. cochenillifera* fresh mass were weighed, immersed in 80% ethanol at 65 °C, ground and homogenized in a polytron (IKA Ultra-Turrax® T25 digital) and centrifuged twice for 10 min at 2000× *g*. At each centrifugation, the samples were filtered on filter paper, and the volume of filtrations combined to the largest volume and totaled to 20 mL in a beaker. The alcoholic extract was stored under refrigeration (8 °C), in sealed containers, to quantify the total and reducing soluble sugars. The results were expressed as percentage.

### 2.4.2. Total Carbohydrates

Total carbohydrate was quantified by the phenol–sulfuric method [27] with 250 μL of pipetted *N. cochenillifera* biomass extracts and the addition of 250 μL of 5% phenol solution per glass test tube, sealed with a capacity of 10 mL and vortexed. Afterwards, 1.25 mL of concentrated sulfuric acid was added, and the solution was stirred again. The tubes were kept in a thermostatic bath (30 °C) for 20 min and shaken again and left at room temperature for 30 min. The AST samples were placed in a spectrophotometer (Genesys 10S UV-VIS) at $\lambda = 490$ ηm; the standard curve was produced with 1% sucrose solution, and the AST results were expressed as percentage.

### 2.4.3. Non-Structural Carbohydrates

#### Reducing and Non-Reducing Sugars

The dinitrosalicylic acid (DNS) methodology determined the content of reducing sugars [28]. A total of 500 μL of alcoholic extract containing plant material from *N. cochenillifera* biomass was pipetted into a glass test tube, and 500 μL of DNS were added to each and placed in a thermostatic bath at 100 °C for 5 min. After cooling, 4 mL of distilled water was added to each tube, resulting in the final reaction mixture. The readings were carried out in a spectrophotometer (Genesys 10S UV-VIS) at $\lambda = 540$ μm; the standard curve was produced with a 0.2% fructose solution, and the AR results were expressed in percentage. The non-reducing sugars (NRA) content was adapted and calculated by the difference between the concentration of the total soluble sugars, reducing sugars, alcohol insoluble solids, and the structural carbohydrates (pectin). The NRA results were expressed as a percentage.

#### Alcohol Insoluble Solids

*N. cochenillifera* biomass pellets were dried in a continuous flow oven at 65 °C for 24 h until dry mass stability; they were then macerated in a crucible and weighed on an analytical balance [18]. The alcohol-insoluble solids content was determined by the residue from the extraction of total soluble sugars from the potato pellets. The results were expressed as percentage.

### 2.4.4. Structural Carbohydrate

#### Pectin

Twenty grams of *N. cochenillifera* biomass was diluted in 400 mL of 0.05 N HCl and kept warm for two hours at 80–90 °C. The total volume was controlled by adding distilled water, as volume was lost by evaporation. At the end of the period, the solution was cooled to room temperature and transferred to a 500 mL beaker, completing the volume with distilled water. After homogenization, the solution was filtered with the aid of cotton, and 200 mL was transferred to a 1 L beaker, adding 250 mL of distilled water. With the aid of a pH meter, the solution was neutralized with 1 N NaOH, and then 10 mL of the same excess alkaline solution was added. The mixture was left to rest overnight. The next day, 50 mL of 1 N acetic acid was added and, after 5 min, 25 mL of 1 N calcium chloride solution was added, while stirred constantly. The solution was heated and boiled for 2 min and left to rest for 3 h. The solution was filtered on previously tared filter paper, and the material

retained on the filter paper was washed with boiling distilled water until the filtrate came out free of chlorides (to check, a 1% silver nitrate solution was used). The filter paper was dried in an oven at 105 °C to constant weight and, by mass difference, the pectin content was obtained, expressed as calcium pectate, according to the equation:

$$\%calcium\,pectate = \frac{(weight\ of\ calcium\,pectate) \times 500}{(mL\ of\ the\ filtrate) \times (sample\ weight) \times 100} \tag{1}$$

### 2.4.5. Adsorption kinetics

The aqueous solution with the biomass was filtered after two hours. Ten aliquots of the filtrate were centrifuged at 15,000 rpm for 10 min. The pellet formed was placed in an oven at 70 °C for 40 min and weighed in an analytical balance. Adsorption isotherms were calculated by the Langmuir model:

$$qe = q_0 Ce / 1 + KCe \tag{2}$$

where $qe$ = amount adsorbed per mass of adsorbent (mg g$^{-1}$); $q_0$ = constant related to adsorption energy; $K$ = Langmuir constant (theoretical monolayer concentration capacity) (mL g$^{-1}$); $Ce$ = equilibrium adsorbate concentration.

### 2.4.6. Turbidity

The aqueous solution turbidity was determined by the nephelometric method using a microprocessed turbidimeter (Del Lab–DLM–2000B). The reading was carried out after stirring the solution clockwise and counterclockwise on a stirring table for 1 h (130 rpm) for clay particles' uniform dispersion and rested for 24 h for complete particles' hydration. Results were expressed as nephelometric turbidity unit (NTU).

### 2.4.7. pH and Electric Conductivity

The pH was determined with bench pH (PHS3BW, BEL Engineering®). The aqueous solution electrical conductivity was determined using a conductivity meter (Marconi FCTP. 906), and the results were expressed in μS cm$^{-1}$.

### 2.4.8. Zeta Potential

*Nopalea cochenillifera* fragments (0.05 g) were diluted in 10 mL deionized water, stirred for 5 min, and sonicated for 2 min (50 kHz). The surface electrical potential of the biomass was determined at 25 °C with a detection angle of 173° using the Zetasizer Nano ZS (Malvern, UK). Analyzes were performed in triplicate, and the results were expressed in mV.

### 2.4.9. Total Coliforms Presence

The most probable number technique was used. Here, 25 mL of the filtrate were aseptically removed and diluted successively (0.1, 0.01, and 0.001). Each dilution used three 10 mL sodium lauryl sulfate (LST) tubes with inverted Durham tubes, which were incubated at 35–37 °C for 24 h. Aliquots of the gas-forming tubes in the LST broth were seeded into tubes containing 5 mL of 2% bright green (VB) broth.

### *2.5. Data Analysis*

Data were submitted to a regression analysis. The models were chosen based on the significance of the regression coefficients, using the *t*-test at 1% probability, for the coefficient of determination and the physical and biological phenomena.

## 3. Results

The content of total carbohydrates, non-structural carbohydrates (reducing sugars, non-reducing sugars, and alcohol insoluble solids), and structural carbohydrates (pectin) in *N. cochenillifera* was 66.80, 61.36, and 5.44% of fresh matter, respectively (Table 1).

**Table 1.** Content of total, non-structural (reducing and non-reducing sugars and, alcohol insoluble solids), and structural (pectin) carbohydrates, and the pH of *Nopalea cochenillifera* fresh matter.

| Compounds | Content |
|---|---|
| Total carbohydrates | 66.80 ± 8.60% |
| Non-structural carbohydrates<br>Reducing sugars<br>Non-reducing sugars<br>Alcohol insoluble solids | 1.95 ± 0.33%<br>47.28 ± 2.22%<br>12.13 ± 0.98% |
| Structural carbohydrates<br>Pectin | 5.44 ± 0.03% |
| pH | 4.70 ± 0.03% |

*Nopalea cochenillifera* biomass adsorbed, agglomerated, and sedimented particles suspended in the aqueous solution regardless of the mass used (x = 2.5; α = 0.01). The aqueous solution turbidity was lower with 1 g of *N. cochenillifera* biomass (Figure 1; Figure 2).

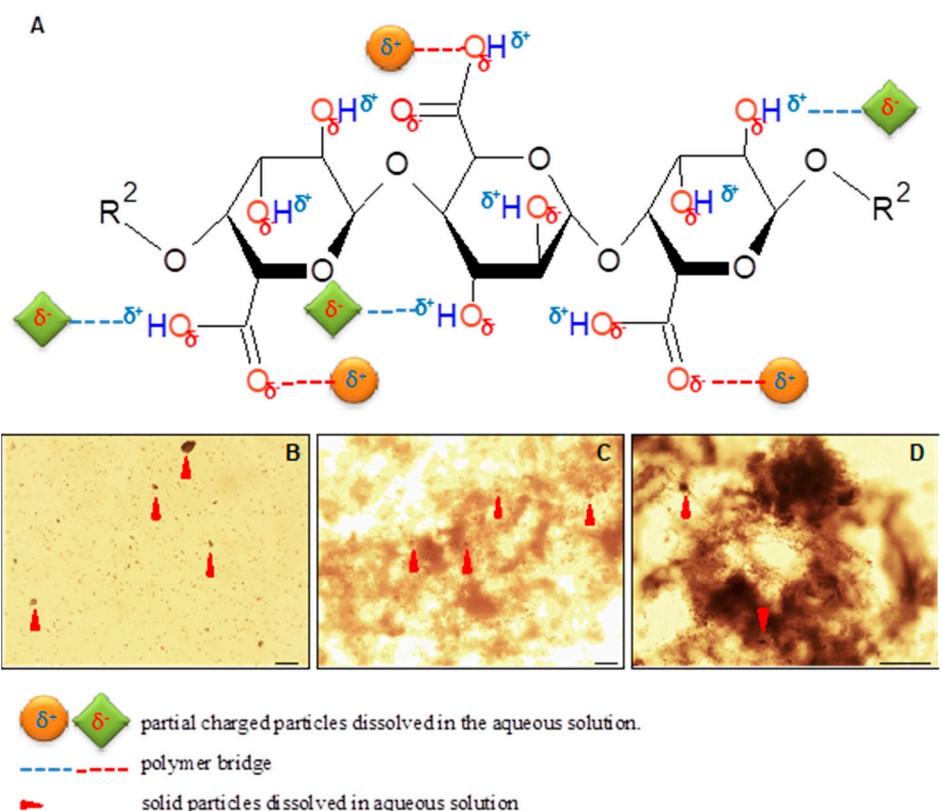

**Figure 1.** Supposed scheme of D-galacturonic acid ionic adsorption, pectin constituent in the mucilage of *N. cochenillifera*, in aqueous solution. The free −COOH and −OH groups with positive and negative partial charges adsorb and aggregate the anionic and cationic polymers dissolved in the aqueous solution by the hydrogen bonds formation (**A**). 10× aqueous solution micrographs without (**B**) and with *N. cochenillifera* at 10× (**C**) and 100× (**D**) forming the aggregates.

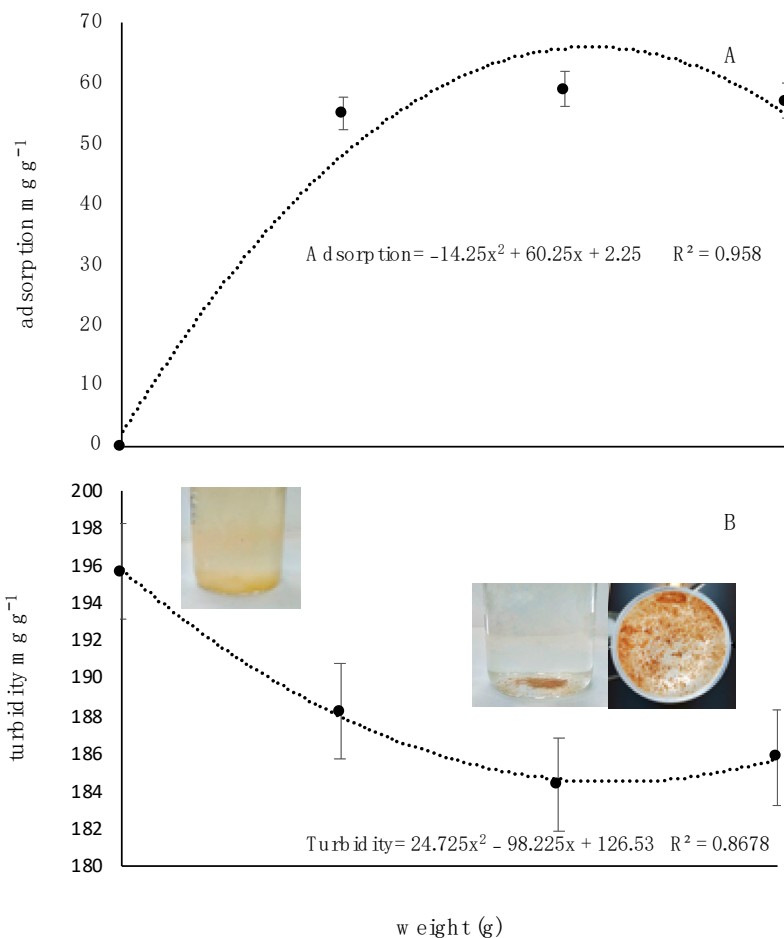

**Figure 2.** Particles adsorption (**A**) and turbidity (**B**) from water treated with *Nopalea cochenillifera* biomass.

The aqueous solution pH and electrical conductivity (Ce) decreased with *N. cochenillifera* addition; however, they did not differ among the different amounts of biomass (pH = 7.1; C e = 186.10; $\alpha$ = 0.01) (Figure 3). Zeta potential (ZP) decreased with the *N. cochenillifera* addition; however, they did not differ among the different amounts of biomass (ZP = 11.0; $\alpha$ = 0.01). At pH < 7.4 the bioadsorbent was negatively charged (Figure 4).

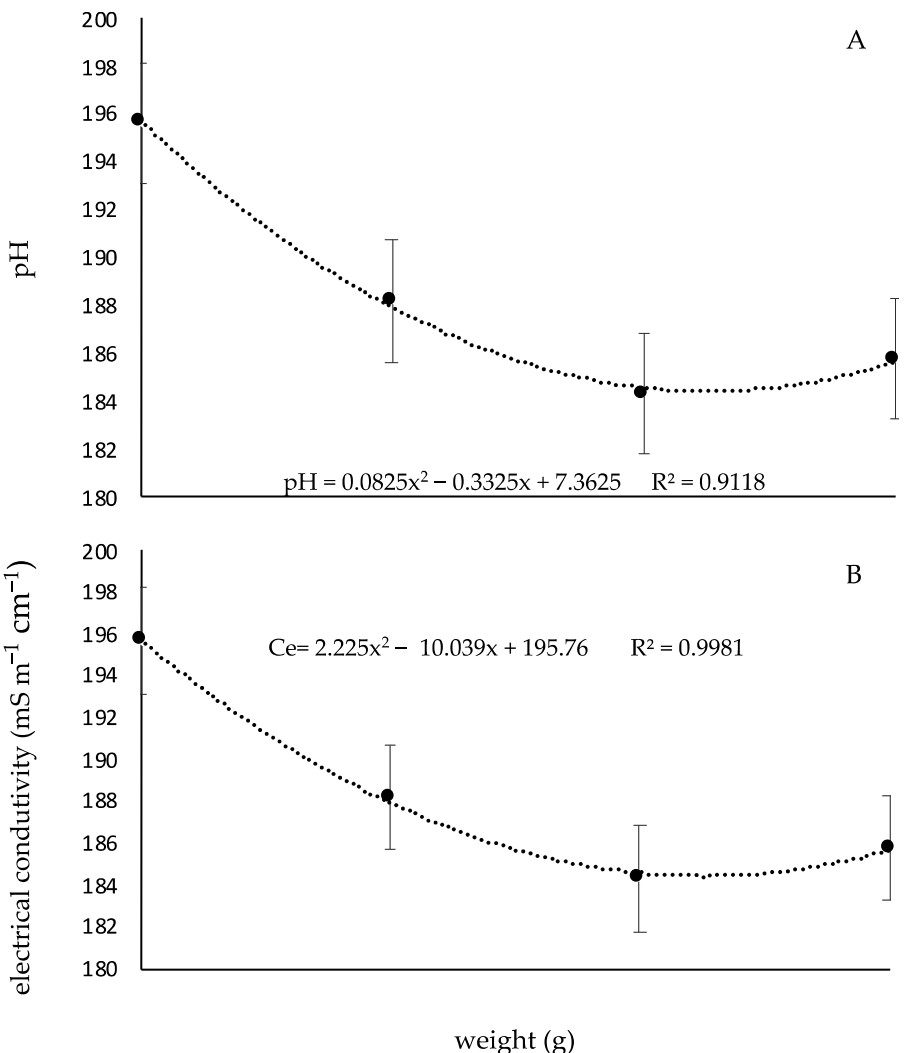

**Figure 3.** Water pH/[H$^+$] (**A**) and electrical conductivity (**B**) treated with *Nopalea cochenillif-era* biomass.

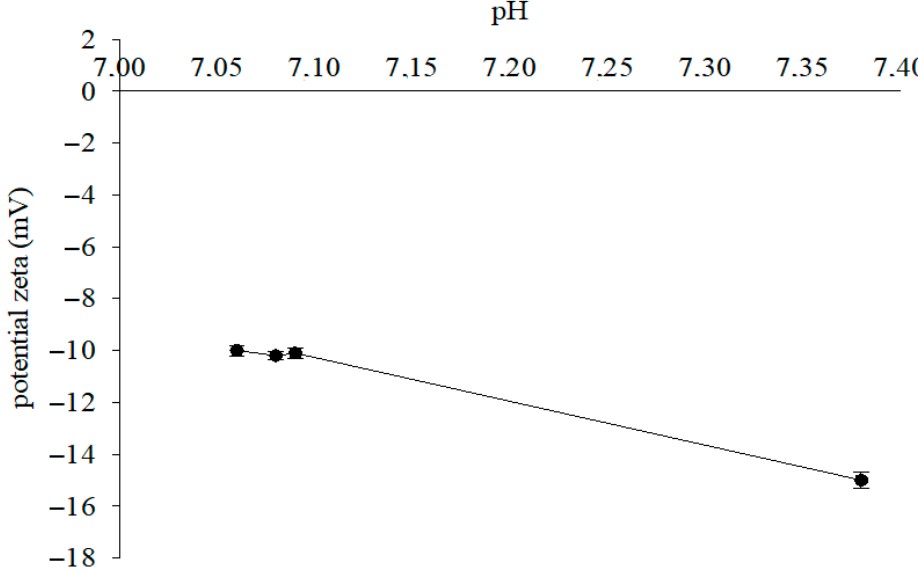

**Figure 4.** *Nopalea cochenillifera* biomass zeta potential as a pH function.

## 4. Discussion

*Nopalea cochenillifera* biomass adsorbed the suspended particles in the aqueous solution, making it more translucent. Adsorption is the process of transferring one or more constituents (adsorbates) from a fluid (adsorptive) phase to a solid surface (adsorbent) phase. In the adsorption process, the molecules present in the fluid phase are attracted to the interfacial zone due to the existence of unbalanced attractive forces on the adsorbent surface [26].

The suspended particles' adsorption in water is explained by the polymer bridge establishment between the suspended particles and the polymeric coagulants present in *N. cochenillifera* [27], specifically polysaccharides and their derivatives, highlighting the high content of total and non-structural carbohydrates (Table 1). Polysaccharides and their derivatives are a complex and heterogeneous ion exchange matrix that adsorbs dissolved particles efficiently, at low cost [18,28] and low contaminant risk. Mucilage, produced in chlorenchyma and parenchyma cells, is the predominantly highly branched polysaccharide in the palm [29]. Pectin, a polysaccharide present in large amount in *N. cochenillifera* (Table 1) and in others such as cactus mucilage [30], is composed of acid D-galacturonic units joined in chains by $\alpha$-(1→4) glycosidic bonds [31]. Pectin-free carboxylic groups with positive and negative partial charges may have adsorbed and aggregated the anionic and cationic polymers dissolved in the aqueous solution by the hydrogen bonds formation [32–34] or dipole and ion–dipole interactions [18], resulting in particle coagulation and, consequently, reduction in aqueous solution turbidity. The presence of numerous hydroxyl groups along the polysaccharide chain increases the intermolecular interaction points. Surface functional groups such as −COOH and −OH were probably responsible for the cationic particles' adsorption being more effectively than the anion ones. Anion extraction is much more difficult than extraction of other dissolved particles, and there are few reports on anion extraction from water using plant biomass [35]. *Opuntia* spp. galacturonic acid individually reduced 30–50% of water turbidity [18]. Anionic polyelectrolytes, lipids, carbohydrates, and alkaloids in *S. potatorum* seed extract destabilize particles in water and increase coagulation capacity [18]. *Melocactus* sp., *O. dillenii*, *Stenocereus griseus*, *Cereus forbesii*, *Aloe arborescens*, *A. vera*, and *Cicer arietinum* flocculate particles smaller than 0.2 mm which generally cannot be separated by natural sedimentation [36].

The aqueous solution pH and Ce decrease after treatment with *N. cochenillifera* biomass and can be generally explained by the acidic nature of the plant biomass (±4.70) and also by the dissolved solids' chelation by polysaccharides and polysaccharide derivatives from the biomass, such as starch (Table 1). These form a complex and heterogeneous ion exchange matrix [37] due to the molecular configuration (acid and basic functional groups) and the properties of biopolymers that adsorb and form complexes with another equivalent number of ions, making them inactive [38]. The reduction of the pH of water treated with O. ficus-indica [39] and *A. vera* leaf gel [37] occurred in a similar way to that described in this research. In these studies, the initial pH of the aqueous solution was neutral or basic because according to the literature, the flocculant activity of cactus biomass is better at neutral or basic pH (Taa et al. 2016). Turbidity and removal of dissolved particles was higher at pH 8.5 [39], and at pH values below or above this value, it caused floating particles, foaming, and the formation of black layers above the beakers. This can be proven by reducing the zeta potential at pH < 7.4. The reduction in turbidity at neutral or basic pH is due to the adsorption of anionic particles dissolved in the aqueous solution to the positively charged biomass surfaces due to the glycosidic bond of the polysaccharide chains. Therefore, flocculant activity and water turbidity reduction by palm biomass are better at neutral or basic pH [38].

Water turbidity was reduced with *O. ficus-indica* biomass due to reduced zeta potential and the protonation of acid groups from polysaccharide chains, increasing intermolecular interaction and causing macromolecule aggregation in aqueous solution [40]. The net negative charge in mildly alkaline neutral pH polysaccharides causes electrostatic repulsion between macromolecules, resulting in smaller average hydrodynamic diameter [41]. In

addition, smaller particles provide more specific surface area to interact with pollutants in solution, allowing for maximum turbidity removal [42,43]. Pectin and lignin, which make up much of the *N. cochenillifera* biomass, contain large amounts of hydroxyl groups with the property of aliphatic and aromatic compounds, but with short chains. Xylan and starch molecules also have large numbers of hydroxyl groups and long helical chains or random coils [44]. Thus, the colloidal particles charges are neutralized as they bind to the clusters of these biomolecules [45]. Therefore, the Ce and pH reduction indicate that dissolved organic and inorganic compounds were complexed and their charges neutralized. The pH, Ce, and zeta potential similarity of the aqueous solution between the *N. cochenillifera* amount indicate that the ion exchange matrix contained in 1 g of biomass was sufficient to adsorb the dissolved solids in the solution. The heterogeneity of the biopolymer matrix is more important than the actual biopolymer amount [37]. Maintaining the pH of the aqueous solution associated with biodegradability, ease of access, and low cost are considered criteria for choosing biomass as a coagulant [19,20]. Several studies indicate that the addition of a metallic coagulant to the biomass of O. ficus-indica and *M. oleifera* reduces water turbidity; however, the advantage in our study is that it only considers plant biomass, with similar action and great advantage in terms of degradability [46,47].

Total coliforms were present in all aqueous solutions. The *N. cochenillifera* biomass addition did not interfere with the total coliforms present in water, the consumption of which can cause deaths related to gastrointestinal diseases such as diarrhea, dysentery, cholera, and typhoid [48]. Therefore, the pretreatment of water from wells, barriers, cisterns, and dams with *N. cochenillifera* is not sufficient to make it safe for human consumption. The consequences to human health due to pathogen exposure are not the same for all individuals or populations. Not all infected individuals develop symptomatic disease. Continued exposure to a pathogen may induce immunity (may be lifelong or temporary), reducing the symptoms' severity. However, children, young people, the elderly, pregnant women, and immunocompromised women are vulnerable to disease, and symptoms can be so severe as to lead them to death [49].

Our research sought alternatives for water pretreatment for populations underserved by public water supply and sanitation policies. Therefore, we reiterate the need for other equally accessible actions to ensure the health of the population, such as chemical disinfection using 0.5 and 1% sodium hypochlorite solution [50]; filtration; solar disinfection, which eliminates microorganisms by exposure to water in dark or opaque containers; solar radiation; and thermal disinfection, whose main mechanism for the destruction of microbes in water is boiling and heating temperatures until pasteurization (>63 °C per 30 min) with natural cooling [51–53].

## 5. Conclusions

*Nopalea cochenillifera* biomass treatment reduces suspended dissolved particles and turbidity of the aqueous solution but needs to be associated with another form of treatment to eliminate heat-trapping coliforms and to ensure safe water for human consumption.

**Author Contributions:** W.S.R., F.B.d.C. and K.P.L. designed the research; V.R.d.N.L., L.d.S.B., Á.G.F.d.S., M.A.R.L., G.N.B.S. and R.R.P.C., performed the experiments; W.S.R., R.d.L.A.B., N.R.d.S.J., P.A.d.S. and A.P.d.A. wrote the manuscript. All authors have read and agreed to the published version of the manuscript.

**Funding:** This research was funded by "Conselho Nacional de Desenvolvimento Científico e Tecnológico (CNPq), Coordenação de Aperfeiçoamento de Pessoal de Nível Superior (CAPES), Fundação de Amparo à Pesquisa do Estado de Minas Gerais (FAPEMIG), Programa de Educação Tutorial (PET) and, Programa de Pós-graduação em Fitotecnia da Universidade Federal de Viçosa for financial support".

**Institutional Review Board Statement:** Not applicable.

**Informed Consent Statement:** Not applicable.

**Data Availability Statement:** Not applicable.

**Acknowledgments:** To Conselho Nacional de Desenvolvimento Científico e Tecnológico (CNPq), Coordenação de Aperfeiçoamento de Pessoal de Nível Superior (CAPES), Fundação de Amparo à Pesquisa do Estado de Minas Gerais (FAPEMIG), Programa de Pós-graduação em Fitotecnia da Universidade Federal de Viçosa and, Programa de Educação Tutorial (PET) for financial support.

**Conflicts of Interest:** The authors declare no conflict of interest.

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
