# Peer review of "Nopalea cochenillifera Biomass as Bioadsorbent in Water Purification"

_water, doi:10.3390/w13152012_

Round 1

Reviewer 1 Report

water-1277244

This manuscript discussed the water purification ability of bio-adsorbent using Nopalea Cochenillifera biomass. Some novelty can be presented by treating real water body with biomass.

However, the result lacks necessary discussion and explanation. Here are my detailed comments:

  1. Too much background was introduced in the Introduction Section. What are the advantages of Nopalea Cochenillifera biomass when comparing with other water purification methods? With my understanding, Nopalea Cochenillifera biomass can be seen as the example of natural polymers. The authors should introduce more examples of natural polymers. What are the unique features of Nopalea Cochenillifera biomass, as compared with other natural polymers? Is it possible to use synthetic polymers for water purification?
  2. The authors should discuss more about the regeneration of Nopalea Cochenillifera biomass after the adsorption process.
  3. What is the pH range of application for Nopalea Cochenillifera biomass? It seems only neutral environment can be suitable for Nopalea Cochenillifera

Author Response

We appreciate the corrections from the reviewers. We revised our manuscript “Nopalea cochenillifera biomass as bioadsorbent in water purification” in accordance with reviewer comments. We are resubmitting the titled manuscript with the answers attached. 

Reviewer 2 Report

Lima et al. have examined the potential application of Nopalea cochenilifera biomass for water treatment.

The adsorption kinetics and turbidity have been measured as a function of biomass dose.

I think the results in this research include some good information; however, it lacks a practical experimental design. I suggest a major revision.

1- The conditions for the turbidity measurements should be described in more detail.

2- The authors assume a linear behavior for turbidity and adsorption between 0 and 1 g of the biomass. This should be verified by testing at concentrations below 1 g.

3- The results should be compared with a reference sample, e.g., a commercial coagulant.

4- The adsorption mechanism described by the authors is unclear.

5- The manuscript should be carefully checked for typographical errors.

Author Response

(The authors gave the same response as above.)

Round 2

Reviewer 2 Report

The authors have not addressed my previous comments. Therefore, I cannot recommend publication.

Author Response

Editorial Staff of Water

July 13, 2021.

          We welcome your suggestions and urge the reviewer to reconsider their position. We've made all suggested corrections, including adding data to the manuscript. We appreciate your attention.            

          We are resubmitting the manuscript entitled: “Nopalea cochenillifera biomass as bioadsorbent in water purification”. We, also, apologize for not following the review guidelines.

            We revise our manuscript follow the reviewer comments. The answers are as follows:

Review Report

1) The conditions for the turbidity measurements should be described in more detail.

We are grateful for the contribution and we described in more detail as can be observed in the lines: 220-228.

2) The authors assume a linear behavior for turbidity and adsorption between 0 and 1 g of the biomass. This should be verified by testing at concentrations below 1 g.

We appreciate your contribution. We submit the data to regression analysis and estimate the range between 0 and 1g. We would also like to clarify that we did preliminary tests with other biomass amounts, but we did not get good results. Furthermore, the entire bibliography consulted makes it clear that small biomass amounts have no effect.

3) The results should be compared with a reference sample, e.g., a commercial coagulant.

We appreciate your contribution but, in our research, it is not our objective to demonstrate that the treatment is better or worse than the conventional one. This comparison would mischaracterize our research. We are not proposing a replacement, but an alternative in serious cases of lack of water treatment. Therefore, we just compare it to the dirty water that is used to give water to humans and animals.

4) The adsorption mechanism described by the authors is unclear.

The mechanism is actually not clear. We have proposed a schematic model in figure 1 and we seek to discuss it throughout the discussion. Our results and discussion are completely in agreement with the bibliography that supports it.

However, to fulfill your suggestion we have added more data related to structural and non-structural carbohydrates. They are directly related to adsorption events. These data can be found in table 1 (lines 268-269) and are described in lines 151-210.

5) The manuscript should be carefully checked for typographical errors.

Thank you very much for the suggestion. We, once more, read carefully by specialized people.

Thank you for your attention.

Round 3

Reviewer 2 Report

My comments are mostly addressed. I recommend publication in present form.